# Fish community composition in the tropical archipelago of São Tomé and Príncipe

**Guillermo Porriños**[1,2,3]*, **Kristian Metcalfe**[3], **Ana Nuno**[3,4], **Manuel da Graça**[5], **Katy Walker**[2], **Adam Dixon**[2,3], **Márcio Guedes**[6], **Lodney Nazaré**[6], **Albertino dos Santos**[7], **Liliana P. Colman**[3], **Jemima Dimbleby**[8], **Marta Garcia-Doce**[9], **Annette C. Broderick**[3], **Brendan J. Godley**[3], **Tiago Capela Lourenço**[1], **Luisa Madruga**[2,5], **Hugulay Albuquerque Maia**[10], **Berry Mulligan**[2], **Philip D. Doherty**[3]

**1** cE3c - Centre for Ecology, Evolution and Environmental Changes & CHANGE - Global Change and Sustainability Institute, Faculdade de Ciências, Universidade de Lisboa, Lisboa, Portugal, **2** Fauna & Flora, Cambridge, United Kingdom, **3** Centre for Ecology and Conservation, Faculty of Environment, Science and Economy, University of Exeter, Penryn, Cornwall, United Kingdom, **4** Interdisciplinary Centre of Social Sciences (CICS.NOVA), School of Social Sciences and Humanities (NOVA FCSH), NOVA University Lisbon, Lisboa, Portugal, **5** Fundação Príncipe, Santo António, Príncipe, São Tomé and Príncipe **6** Oikos–Cooperação e Desenvolvimento, Água Grande, São Tomé and Príncipe, **7** ONG MARAPA, São Tomé, São Tome and Príncipe, **8** School of Biological and Marine Sciences, University of Plymouth, Plymouth, United Kingdom, **9** University of La Laguna, Tenerife, Spain, **10** Universidade de São Tomé, São Tomé, São Tomé e Príncipe

* gporrinos@alunos.ciencias.ulisboa.pt

## Abstract

Understanding species distribution across habitats and environmental variables is important to inform area-based management. However, observational data are often lacking, particularly from developing countries, hindering effective conservation design. One such data-poor area is the Gulf of Guinea, an understudied and biodiverse region where coastal waters play a critical role in coastal livelihoods. Here, we describe the results of the largest national-scale Baited Remote Underwater Video Systems (BRUVS) survey in the region, aiming to understand the effects of several environmental variables on fish community composition and diversity. From 2018 to 2020, we successfully deployed 417 benthic BRUVS in the coastal waters of the São Tomé and Príncipe (STP) archipelago. Species richness and relative abundance were higher in deeper waters, on steeper slopes, and in rocky reef habitats. Nevertheless, maerl and sand habitats also hosted unique, and economically important species. Our results potentially indicate historical impacts of fishing in the archipelago, especially in São Tomé Island, where observed fishing effort is higher. Indeed, abundance of large predatory fish was low in both islands and abundance of species targeted by artisanal fisheries was lower in São Tomé than in Príncipe. Our results provide crucial information supporting the designation and future monitoring of marine protected areas in STP.

**Data Availability Statement:** Data and code used for analyses in this study are available at https://doi.org/10.5281/zenodo.12790789

**Funding:** Fieldwork described here was undertaken within the project "Establishing a network of

marine protected areas across São Tomé and Príncipe through a co-management approach" and was funded by Blue Action Fund and Arcadia Fund for Nature - a charitable fund of Lisbet Rausing and Peter Baldwin. Preliminary work was funded by the Darwin Initiative - a UK government grant scheme (Project 23–012), Forever Príncipe and Halpin Trust. G.P. acknowledges funding from Fundação para a Ciência e Tecnologia (FCT) doctoral grant nº UI/BD/151263/2021. A.N. acknowledges the support of the European Union's Horizon 2020 research and innovation programme under the Marie Skłodowska-Curie grant agreement SocioEcoFrontiers No. 843865. K.M. acknowledges funding support from the Darwin Initiative (Project 23-011 and 26-014), Waterloo Foundation, and the Wildlife Conservation Society Congo Programme. G.P and T.C.L. acknowledge the support received from cE3c - Center for Ecology, Evolution and Environmental Changes through FCT's strategic project UIDB/00329/2020 (doi: 10.54499/UIDB/00329/2020). The funders had no role in study design, data collection and analysis, decision to publish, or preparation of the manuscript.

**Competing interests:** The authors have declared that no competing interests exist.

## Introduction

Marine ecosystems provide numerous services and play a critical role in supporting livelihoods of coastal human communities [1]. Coastal zones, which comprise 11% of the total ocean surface, account for 90% of global fisheries catches, and support over one third of the global human population [2]. However, anthropogenic activities in recent decades have dramatically impacted biodiversity and ecosystem health and reduced the capacity of these systems to provide such services [3]. In response, Marine Protected Areas (MPAs), have been increasingly advocated as a means to improve the spatial and temporal allocation of human activities, and mitigate their impact in the marine environment [4]. If managed effectively, MPAs have the potential to restore population size and diversity within their boundaries and support neighbouring fisheries through spill-over and larval export (e.g. [5]).

Effective design of area-based management measures, including MPAs, requires an understanding of marine ecological processes and distribution of marine biodiversity [6]. Habitat classes (e.g. rocky reefs, sand, and maerl), as well as other environmental variables (e.g. depth) have often been used in MPA design as proxies for species diversity and distribution, under the premise that different habitat classes and abiotic characteristics support different biological communities [7]. By representing diverse environmental conditions within the boundaries of a marine reserve, it is assumed that ecological processes and biodiversity are also implicitly represented [8]. However, these parameters alone may be insufficient to capture biological patterns, and data collected through biodiversity assessments are therefore needed to determine whether environmental variables can reliably predict species occurrence and abundance [8, 9]. This consequently requires robust ecological monitoring to establish biological benchmarks and assess change over time and space.

Common techniques used to monitor and assess the health and status of ecological coastal communities include Underwater Visual Census (UVC), scientific fishing, and Baited Remote Underwater Video Systems (BRUVS), amongst others [10]. BRUVS are increasingly being used to assess mobile assemblages, as they can be relatively low-cost and are non-invasive [11]. Whilst BRUVS have been deployed across temperate and tropical systems, to our knowledge there have been no large scale BRUVS surveys conducted along the Atlantic coast of Africa, where the lack of baseline ecological data is considered a major barrier to designing effective conservation strategies [12].

One such area that has been poorly assessed is the Gulf of Guinea Large Marine Ecosystem, a marine biodiversity hotspot [13], with a high diversity of marine species [14] and reef fish endemism [15]. To address this critical data gap, we conducted the first comprehensive national-scale BRUVS survey in São Tomé and Príncipe (Fig 1), a small insular state where marine resources play a critical role in coastal livelihoods. The aim of this study was to characterise STP's marine environment, and, specifically: (1) characterise species composition of marine fish communities; (2) ascertain if selected environmental variables (habitat, depth, island, slope, distance to shore, or season) affect diversity indicators (species richness, relative abundance, and evenness); and (3) determine if community composition was different across these environmental variables and identify which species are driving differences among communities. These data were used to inform an MPA co-design process involving coastal communities, the government and the private sector [16]. Additionally, these data will provide a benchmark against which to assess long-term management effectiveness, a key component of area-based management success.

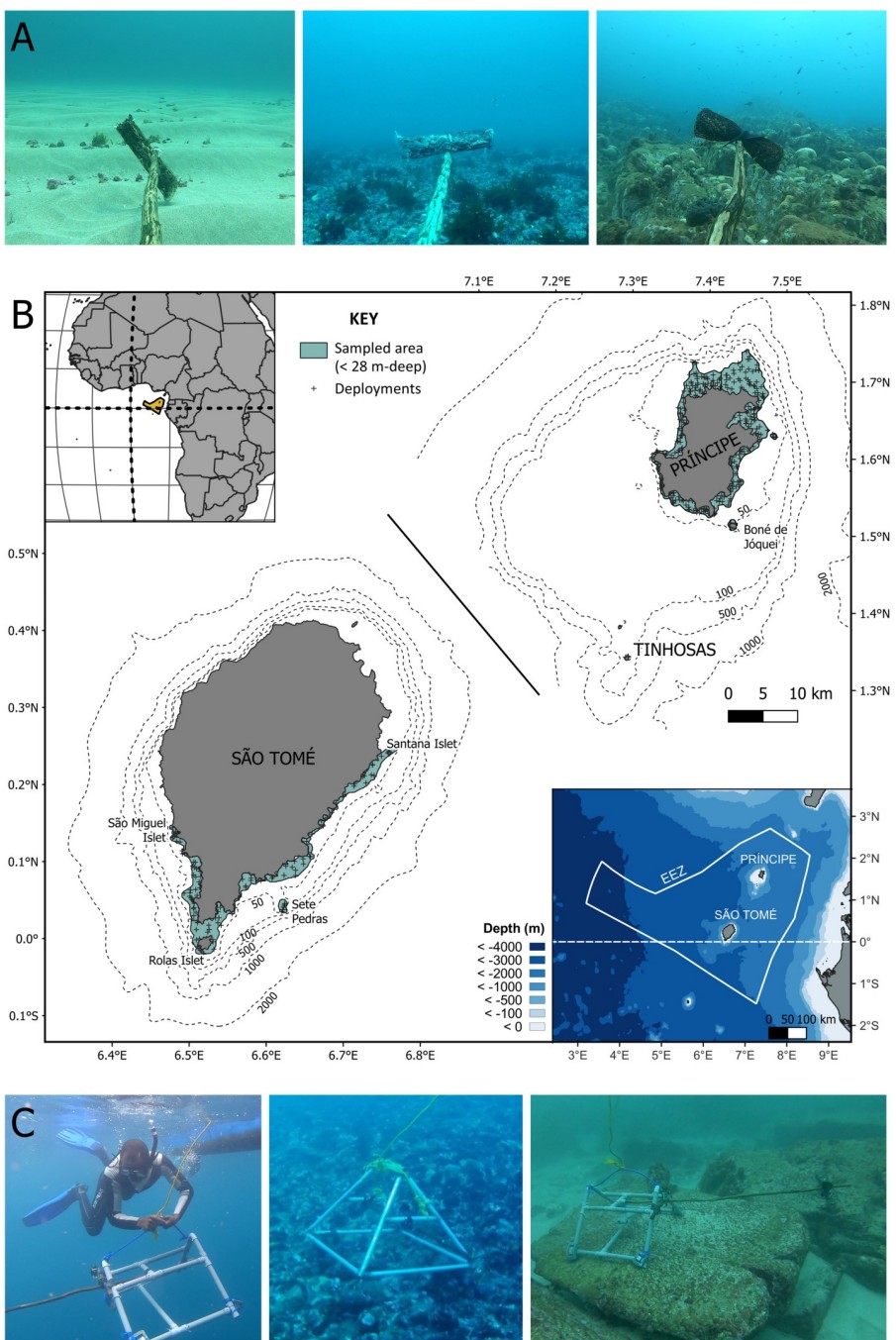

**Fig 1. Habitat, location of São Tomé and Príncipe and pictures of BRUVS fieldwork.** A) Marine habitats considered in this study: sand (left), maerl (centre) and rocky reefs (right)–extracted from BRUVS footage (© Fauna & Flora, Fundação Príncipe, Oikos, MARAPA). B) Top left: Location of São Tomé and Príncipe, Bottom right: Bathymetric map of the Gulf of Guinea denoting STP's Exclusive Economic Zone (solid white line), and Centre: BRUVs deployment locations and sampling area illustrated. (© Bathymetric data: GEBCO and digitised nautical charts, see Supplementary Materials). C) *In situ* photographs of BRUV deployments (left), São Tomé BRUVS (centre) and Príncipe's BRUVS (right) (© G. Porriños).

## Material and methods

### Study site

São Tomé and Príncipe (officially the Democratic Republic of São Tomé and Príncipe, hereafter referred to as STP) is an archipelago in the Gulf of Guinea (Central Africa), comprised of two oceanic islands of volcanic origin (São Tomé and Príncipe) and several smaller islets (Fig 1). The islands have a steep underwater relief that results in a relatively small shelf platform (Fig 1). The larger of the two main inhabited islands - São Tomé (865 km$^2$, 189,423 inhabitants) is located 280 km west from Gabon and has a narrow insular shelf (485km$^2$). In contrast, Príncipe (136 km$^2$, 8,277 inhabitants), located 150 km north-east of São Tomé and 240 km west of Equatorial Guinea [17], has a larger insular shelf (1,085km$^2$) that extends to two small islets in the south called the *Tinhosas*, a remote and important area for seabirds [18].

The Gulf of Guinea has large fluctuations of Sea Surface Temperature (SST), with colder water (mean SST < 23º C) between July and October due to strong seasonal upwelling and warmer water (mean SST > 28º C) between November and May [19]. This seasonal influx of nutrient-rich waters contributes to some of the most productive coastal and offshore waters in the world, which are rich in fishery resources and play a critical role in supporting local livelihoods [20]. Indeed, STP is considered amongst the world's most fishery-dependent countries [21], with fish contributing to 50% of the animal protein consumed in the archipelago and 8% of the active population directly involved in the artisanal fisheries sector [22]. Artisanal fishing is open access, and the country has low capacity to monitor foreign industrial fishing [23]. Artisanal fisheries are dominated by hook-and-line and surface gillnet fishing, with seine nets and demersal gillnets used by a small proportion of vessels [22, 24]. Artisanal fishers have reported a decline in fish catches [25–27] and biological surveys indicate that fish populations might have been impacted by long-term fishing effort [28, 29].

### BRUVS sampling design

Sampling area extended from the coastline out to the 28-metre isobath. On Príncipe, the sampling area was 100 km$^2$ and encompassed all coastal areas (Figs 1B and S1). On São Tomé, only the Southern half of the island, covering 70 km$^2$, was sampled due to logistical constrains (Figs 1B and S2). The Tinhosas islets were also sampled, with a sampling area of 0.08 km$^2$ from the coastline out to the 28-m isobath. Although located within Príncipe's shelf, they were considered a separate island category due to their remoteness and unique characteristics (Figs 1B and S1).

Sampling was conducted for two years in Príncipe (2018/2019 and 2019/2020) and one year in São Tomé (2019/2020). Sampling was stratified across two seasons, with two sampling rounds per year corresponding to (1) the colder-water "gravana" season (June to September) and (2) the warmer-water "summer" season (December to May). BRUVS deployment locations were randomly sampled with a density of one point per ~1.6 km$^2$ per sampling round, resulting in 40 and 60 points per sampling round on São Tomé and Príncipe, respectively (S1 and S2 Figs). Random sampling was used since the lack of habitat distribution information prevented a habitat-stratified sampling design. Random points were generated using QGIS (v. 3.28.1) [30] with a minimum distance of 400 m between deployments to avoid overlapping bait plumes and ensure independence of sampling [11]. In 2018/2019, this approach resulted in a low representation of rocky reefs due to the smaller extent of this habitat, therefore, during 2019/2020 sampling, 70 and 39 additional BRUVS were deployed, targeting rocky reef areas of São Tomé and Príncipe, respectively (S1 and S2 Figs). Location of rocky reefs was determined *a priori* with the help of local fishers. Finally, the Tinhosas Islets were sampled opportunistically, with three points sampled three different times from 2018 to 2020 (S1 Table). See

Supplementary Information for further methodological details and S1 Table for a summary of deployments across islands, sampling rounds and habitats.

## BRUVS data collection

BRUVS frames were constructed from PVC pipes (Príncipe) or welded galvanised steel (São Tomé) and weighted for stability on the seafloor (Fig 1C). Underwater cameras (GoPro Hero 5 and 7) in an underwater casing were attached to the frame 40 cm above the seafloor, with a protruding bait arm with a bait cage filled with 500 g of chopped *Euthynnus alleteratus* and *Auxis thazard*, locally known as *fulu fulu*. This was considered an appropriate bait choice as it is oily, available locally all year and persistent in the bait cage [11] and one of the most common bait types used by artisanal fishers in São Tomé and Príncipe [24].

BRUVS videos were analysed for 60 minutes, as this has been shown to be effective at detecting 90% of species richness in rocky reef environments [11]. Thus, BRUVS were deployed for a minimum of 70 minutes, creating a five-minute time buffer around the video to minimise interference from the boat, deployment, or hauling. Sampling was conducted during daylight between 09:00 and 15:00 recording depth for each deployment, measured to the nearest 10 cm using a handheld sonar (model: *Plastimo EchoTest II*).

## Video analysis

Videos were analysed using VLC Media Player (v.2.1.3, www.videolan.org/vlc) by five trained observers. All teleost and elasmobranch species observed in the video were recorded and identified to the lowest possible taxonomic level using existing species list for the archipelago [29, 31]. Videos were analysed starting five minutes after the BRUVS had settled on the seabed. For each species, relative abundance, defined as the maximum number of individuals observed per frame (MaxN), was recorded alongside the time in the video at which they were observed [32, 33].

## Variable definition

For each deployment, the following diversity indices were determined: species richness (*S*), species relative abundance (maximum MaxN), deployment relative abundance (Total species MaxN) and $E_{1/D}$ evenness index. Species richness (*S*) corresponds to the number of different fish species observed on each deployment, while deployment relative abundance (Total MaxN) corresponds to the sum of the maximum MaxN value of all species present in a deployment. $E_{1/D}$ evenness index measures how close in number each species in a deployment is and is independent of species richness [34]. $E_{1/D}$ ranges from 0 (low evenness) to 1 (high evenness) and was calculated after Smith & Wilson [34].

Finally, the following independent variables were assigned to each deployment: island (São Tomé, Príncipe, or Tinhosas), habitat (rocky reef, sand, or maerl), depth (metres), season (ordinal date), distance to shore (metres), and seabed slope (degrees). Dominant habitat type at each deployment was qualitatively described from the field of view of BRUV recordings, and were classified as (1) sandy bottoms, (2) maerl or rodolith beds (unattached, calcareous algae that form globular structures on sandy bottoms) and (3) rocky reefs [35]. Seabed slope was used as a measure of seascape topographic complexity and derived from bathymetric data using the R package *raster* [36]. Distance to shore was computed using the R package *geosphere* [37]. See Supplementary Information for further methodological details.

## Statistical analysis

To understand the effect of environmental and physical variables (habitat type, island, seabed depth, seabed slope, distance to shore, and season) on diversity indicators (species richness, relative abundance, and evenness) we used Generalised Additive Models (GAMs). Island and habitat were fitted as fixed categorical effects, excluding the Tinhosas islets due to their small sample size (n = 6); whilst depth, slope, and distance to shore were fitted as thin plate splines via restricted maximum likelihood (REML). To model intra-annual (i.e., within-year) seasonality we fitted ordinal date (n = 365) as a cyclic cubic spline via REML, also grouping by island to assess whether seasonal effects held across islands. We used a negative binomial GAM for richness and abundance due to overdispersion and a beta-regression GAM for evenness (values range between 0 and 1). Akaike Information Criterion (AIC) was used to select most parsimonious models across all possible combinations of effects, retaining models with DAIC < 6 and excluding models in which a simpler (nested) model attained stronger weighting [38]. Weighted averages of best performing models were used to estimate effects, substituting by zero when a predictor was not present in a model. The level of support of predictors in the top model set was assessed by summing weighted AICs (Summed Weights, SW) across all models containing the variable of interest (e.g., if, in a top model set, predictor "A" is present in two models with weighted AICs = 0.36 and 0.37, then $SW_A = 0.73$). SW ranges from 0 to 1, with values closer to one indicating a greater and more consistent support for the predictor of interest across the retained models. Additionally, the contribution of predictors to the deviance explained within the individual models in the top model set was calculated as per Lai et al. [39] to account for the average shared variance of predictors with concurvity. See Supplementary Material for further methodological details.

We used two-way Permutational Analysis Of Variance (PERMANOVA) and distance-based Redundancy analysis (dbRDA) on species MaxN data to explore patterns of community composition across environmental variables. For these analyses, data from the Tinhosas Islets was retained, as these methods are robust with unequal sampling [40]. Due to wide distribution and variance in species counts, the data were fourth-root transformed to reduce heterogeneity and the potential effect of high relative abundance values masking potential trends in species and family assemblages (e.g. [41]). To visually explore the effect of environmental variables and estimate the percentage of the variance explained by environmental variables, we used distance-based Redundancy Analysis (dbRDA) with Bray-Curtis distances on fourth root transformed MaxN data and used a permutation test to assess the significance of the resulting ordination (see Supplementary Material for further methodological details). To test whether fish community composition was dissimilar across environmental variables we used PERMANOVA (permutations = 9999) with Bray-Curtis distances on fourth root transformed MaxN data and Benjamini–Hochberg correction for multiple comparisons.

To understand if species were driving dissimilarities across pairs of factors habitats and islands, we used similarity percentages (SIMPER) on Bray-Curtis distances of fourth root transformed fish MaxN data. A permutation test (n = 9999) was performed to identify the species for which the differences among pairs of factors contributed significantly to overall dissimilarities [42]. As SIMPER pairwise comparisons only account for one independent variable at a time, focus was given to rocky reef habitats due to their presence across all three islands.

All statistical analyses were conducted using the statistical software R, version 4.1.2 [43] and the R packages *mgcv* [44], *MuMIn* [45], *gam.hp* [39] and *vegan*, v. 2.6–4 [42] for analyses, and the packages *ggplot2* [46] and *gratia* [47] for producing figures and plots.

## Results

Across 498 BRUVS deployments, 81 were excluded due to data or camera failure (n = 68), low visibility (n = 9) and loss of BRUVS (n = 4). A total of 417 deployments were thus retained for analysis, of which 263 were in Príncipe, 6 in the Tinhosas, and 148 in São Tomé (see Supporting information, S1 and S2 Tables for a summary of deployments and environmental conditions). One BRUVS deployment corresponds to one hour of video.

### Observed species

Across the 263 BRUVS retained for analysis on Príncipe, 116 teleost species (grouped into 40 families), and 9 elasmobranch species (grouped into 4 families) were identified (S3 and S4 Tables). The most species-rich teleost families on Príncipe Island were Carangidae, Serranidae and Labridae (n = 13, 9 and 7 species, respectively), which accounted for 23% of all observed species in Príncipe (S3 and S4 Tables). Carangidae, Serranidae and Labridae were also the most commonly occurring families, occurring in 86% (n = 223), 66% (n = 171), and 52% (n = 134) of Príncipe's deployments, respectively (Fig 2A and S4 Table). The three most observed species were blue runner (*Caranx crysos*), yellow jack (*Carangoides bartholomaei*), and Santomean comber (*Serranus pulcher*), which were observed in 60% (n = 157), 49% (n = 127), and 40% (n = 105) of Príncipe's deployments, respectively (S3 Table). The most abundant family was Serranidae, followed by Carangidae and Pomacentridae (mean MaxN = 13.8, 9.5, and 4.1, respectively; Fig 2B and S4 Table). The most abundant species were Creole fish (*Paranthias furcifer*), brown chromis (*Chromis multinlieata*), and mackerel scad (*Decapterus macarellus*) (mean MaxN = 10.4, 3.2 and 3.0, respectively; S3 Table).

Sharks and rays occurred in 3.0% and 7.2% of Príncipe's deployments, respectively, with a mean MaxN of 0.052 sharks and 0.081 rays per deployment. Shark observations were of black-tip shark (*Carcharhinus limbatus*), lemon shark (*Negaprion brevirostris*), and Galapagos shark (*Carcharhinus galapagensis*) of the family Carcharinidae (species richness = 3, occurrences = 1.5% of deployment, mean MaxN = 0.031) and of the species nurse shark (*Ginglymostoma cirratum*) of the family Ginglymostomidae (species richness = 1, occurrences = 1.1% of deployment, mean MaxN = 0.015). Ray observations were of smalltooth stingray (*Hypanus rudis*) and round stingray (*Taenuriops grabatus*) of the family Dasyatidae (species richness = 2, occurrences = 7% of deployments, mean MaxN = 0.081). There was also an observation of an unidentified devil ray of the family Mobulidae (species richness = 1, occurrences = 0.38% of deployments, mean MaxN = 0.004) (S3 and S4 Tables).

Across the 148 BRUVS deployments retained for analysis on São Tomé, 101 teleost species (41 families), and 4 elasmobranch species (4 families) were identified. The most species-rich teleost families were Carangidae, Serranidae, and Labridae (n = 10, 9 and 6 species, respectively), which collectively accounted for 25% of all observed species (S4 Table). Monacanthidae, Serranidae and Labridae were the most observed families, occurring in 59% (n = 85), 59% (n = 85) and 52% (n = 75) of São Tomé's deployments, respectively (Fig 2C and S4 Table). The most observed species were orangespotted filefish (*Cantherhines pullus*), Creole fish, and Newton's wrasse (*Thalassoma newtoni*), which occurred in 51% (n = 74), 39% (n = 56) and 38% (n = 55) of São Tomé's deployments, respectively (S3 Table). In terms of relative abundance, Serranidae, Pomacentridae, and Labridae (mean MaxN = 28.7, 10.0 and 4.3, respectively) were the most abundant families (Fig 2D and S4 Table); while Creole fish, brown chromis, and orangespotted filefish (mean MaxN = 26.1, 8.6 and 3.1, respectively) were the most abundant species (S3 Table).

Sharks and rays occurred in 4.7% and 3.4% of São Tomé's deployments, respectively, with a mean MaxN of 0.054 sharks and 0.033 rays per deployment. Shark observations were

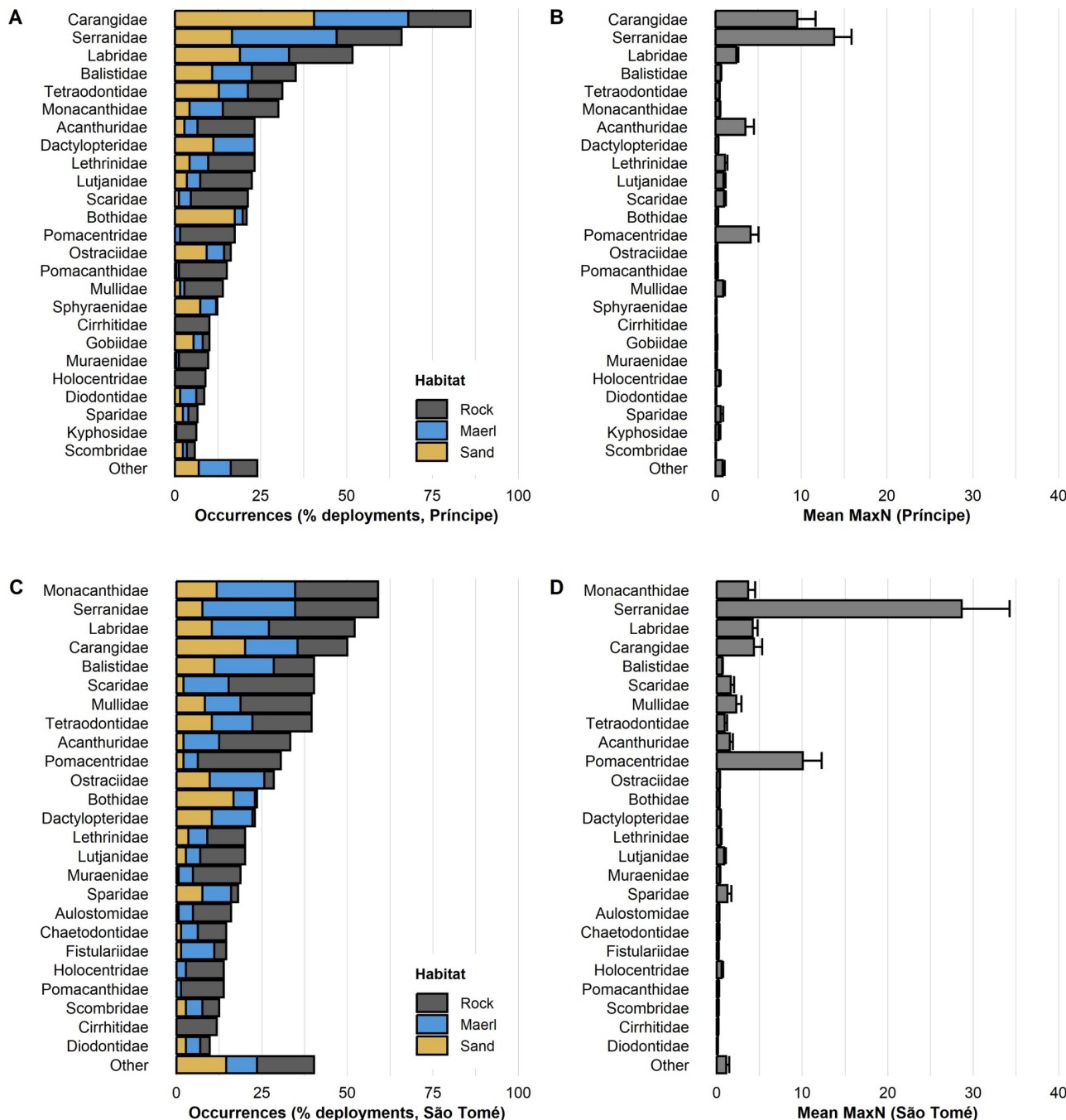

**Fig 2. Family composition of deployments in terms of occurrences and abundance.** Occurrences of the 25 most commonly occurring families in Príncipe (A) and São Tomé (B), represented as % of deployments with colours representing habitat type in relation to all deployments. Relative abundance of these families, represented as mean MaxN with standard error (SE) bars for Príncipe (C) and São Tomé (D). High abundance of Serranidae were caused by large schools of the commonly occurring species *Paranthias furcifer*.

comprised of scalloped hammerhead (*Sphyrna lewini*) of the family Sphyrnidae (species richness = 1, occurrences = 2.0% of deployments, mean MaxN = 0.035) and Galapagos shark of the family Carcharinidae (species richness = 1, occurrences = 0.7% of deployments, mean MaxN = 0.007). Ray observations comprised of smalltooth stingray of the family Dasyatidae

(species richness = 1, occurrences = 2.8% of deployments, mean MaxN = 0.028) and an unidentified devil ray of the family Mobulidae (species richness = 1, occurrences = 0.7% of deployments, mean MaxN = 0.007) (S3 and S4 Tables).

Finally, across the 6 deployments retained for analyses on the Tinhosas Islets (a rocky reef environment), we observed 44 teleost species (22 families) but no elasmobranch species (S3 and S4 Tables). Creole fish, brown chromis, and African red snapper (*Lutjanus agennes*) (mean MaxN = 124.2, 67.3 and 9.5, respectively) were the most abundant species. Black triggerfish (*Melichthys niger*) were observed in 3 of the 6 deployments at the Tinhosas, with relatively high abundances (mean MaxN = 4.6). This species was absent from Príncipe and in São Tomé was only observed in three deployments in the Rolas Islet (Southern São Tomé, Fig 1).

## Species richness and relative abundance

Mean observed species richness across all deployments was 8.9 (SD = 7.6) species per deployment (S5 Table). Habitat and depth strongly influenced trends seen in the top model set (SW = 1), as did slope (SW = 0.73) and season, and its interaction with island (SW = 0.52 and 0.59). Island had less influence (SW = 0.12) and distance to shore was absent from best performing models (Fig 3 and S6 and S7 Tables). Retained models explained 65.2 to 65.9% of the deviance (S8 Table). Habitat had the largest effect on species richness (Fig 3 and S8 Table), contributing to 78.1 to 85.5% of the explained deviance in the models in which it was present. Rocky habitats had the highest number of observed species (mean = 20.2, SD = 5.8) while sandy bottom habitats had the lowest (mean = 4.1, SD = 2.8; Fig 3 and S5 Table). Depth and slope had positive monotonic relationships with species richness, with higher richness observed in deeper waters and steeper seafloors (Fig 3), contributing to 6.1 to 8.6% of the explained deviance in the models in which they were present. Interannual seasonality effects (Fig 3) were only apparent for Príncipe Island, where species richness was highest in April and lowest in August.

Mean observed relative abundance (MaxN) across all deployments was 53.2 (SD = 87.4) per deployment. Habitat, depth, and the interaction between season and island strongly influenced trends seen in the top model set (SW = 1, 0.89 and 0.71, respectively), as well as island, slope, and distance to shore (SW = 0.59, 0.47 and 0.49, respectively), with less influence of season (SW = 0.26) (S6 and S7 Tables). Retained models explained 43.4 to 45.8% of the deviance (S8 Table). Habitat had the largest effect on relative abundance (Fig 3 and S8 Table), contributing to 66.9 to 91.7% of the explained deviance in the models in which it was present. Rocky habitats had the highest relative abundance (mean MaxN = 146.3, SD = 126.4) and sandy bottom habitats the lowest (mean = 14.0, SD = 29.3; Fig 3 and S5 Table). Depth and slope had positive monotonic relationships with relative abundance (i.e., higher abundance in deeper and steeper seafloors, Fig 3). The effect of distance to shore was not linear, showing a peak in richness at the shore and at ~3500 m offshore. Seasonal effects were only apparent for Príncipe, showing a peak in abundance in April and a minimum in October. Island effects were small, contributing to 1.4 to 3.4% of the explained deviance of the models in which it was present, with a slightly higher relative abundance in São Tomé.

Mean observed $E_{1/D}$ evenness index across all habitats and islands was 0.56 (SD = 0.30). Habitat, the interaction of season and island, depth and slope strongly influenced trends seen in the top model set (SW = 1, 0.78, 0.73 and 0.66, respectively), with less influence of season (SW = 0.33). Island and distance to shore were absent from top model set (S6 and S7 Tables). The top model set explained 44.5 to 47.2% of the deviance (S8 Table). Habitat had the largest effect on evenness, contributing to 75.0 to 92.7% of the model deviance (Fig 3 and S8 Table). Evenness was highest on sand (mean = 0.75, SD = 0.23) and lowest on rocky reefs

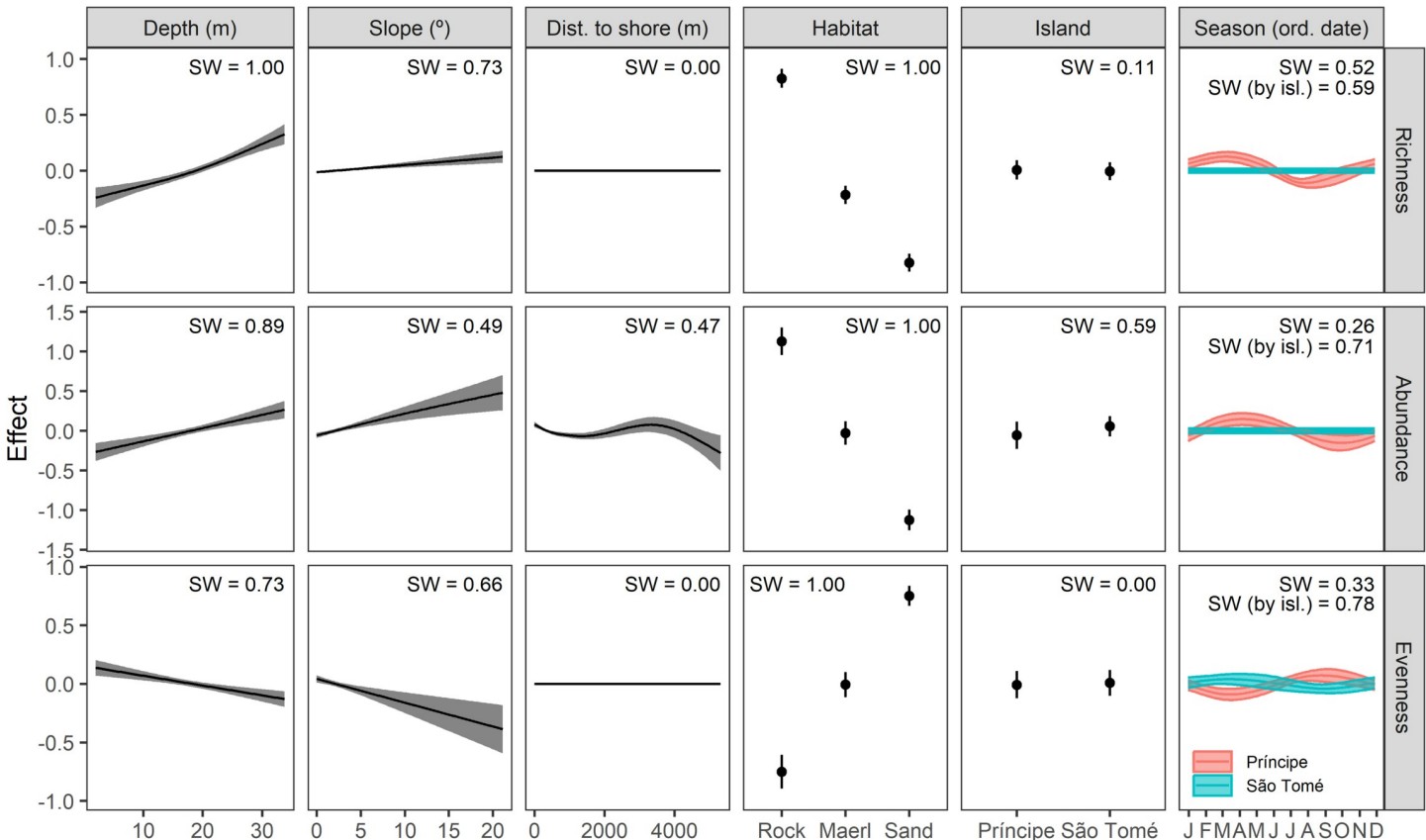

**Fig 3. Partial effect of predictors on diversity indicators.** Predicted values from GAMs described in S6 Table, representing the partial effect of environmental and physical variables (depth, slope, distance to shore, habitat, island and season) on diversity indicators (species richness, S; fish relative abundance, MaxN; and $E_{1/D}$ evenness index). Depth, distance to shore and slope were modelled as thin plate splines, while intra annual seasonality (ordinal date) was modelled as a cyclic cubic spline; months represented in the *x* axis. Seasonality splines were also grouped by island to assess the effects of their interaction. Partial effects of smooth terms were calculated using the R package *gratia* [47], averaging across models retained in the top model set and substituting by zero when a predictor was not present in a model. The level of support of each predictor in the best performing models was calculated as the sum of weighted AICs (Summed Weights, SW); with values closer to one indicating a greater and more consistent support for the predictor of interest across the retained models. A detail of the smooth terms and residuals is available in S4 Fig.

(mean = 0.26, SD = 0.17) (Fig 3 and S5 Table). Depth and slope had a negative, monotonic relationship with evenness (i.e., lower evenness at deeper and steeper seafloors) (Fig 3), and intra-annual seasonality had opposites effects on each island, with lowest and highest evenness in April and September in Príncipe, and vice versa in São Tomé (Fig 3).

## Patterns of community composition

Results from dbRDA showed that the constrained variance (i.e., the variance in species composition explained by environmental variables) comprised 25.6% of the total variance. Permutation tests indicated that the resulting dbRDA ordination (Figs 4 and S5) was significant (df = 8, pseudo-*F* = 18.7, p<0.01), but only the first four constrained axes (comprising 95.5% of the constrained variance) were significant (p<0.01, S9 Table). Constrained axes dbRDA1, dbRDA2 and dbRDA3 (which explained 61.3%, 20.4% and 10.2% of the constrained variance, respectively) showed separation between rocky and non-rocky habitats (dbRDA1), sand and maerl habitats (dbRDA2), and Príncipe and São Tomé Islands (dbRDA3), although with overlap across categories (Fig 4 and S10 Table). PERMANOVA further showed that species composition was significantly affected by habitat, island, depth, slope, distance to shore, and

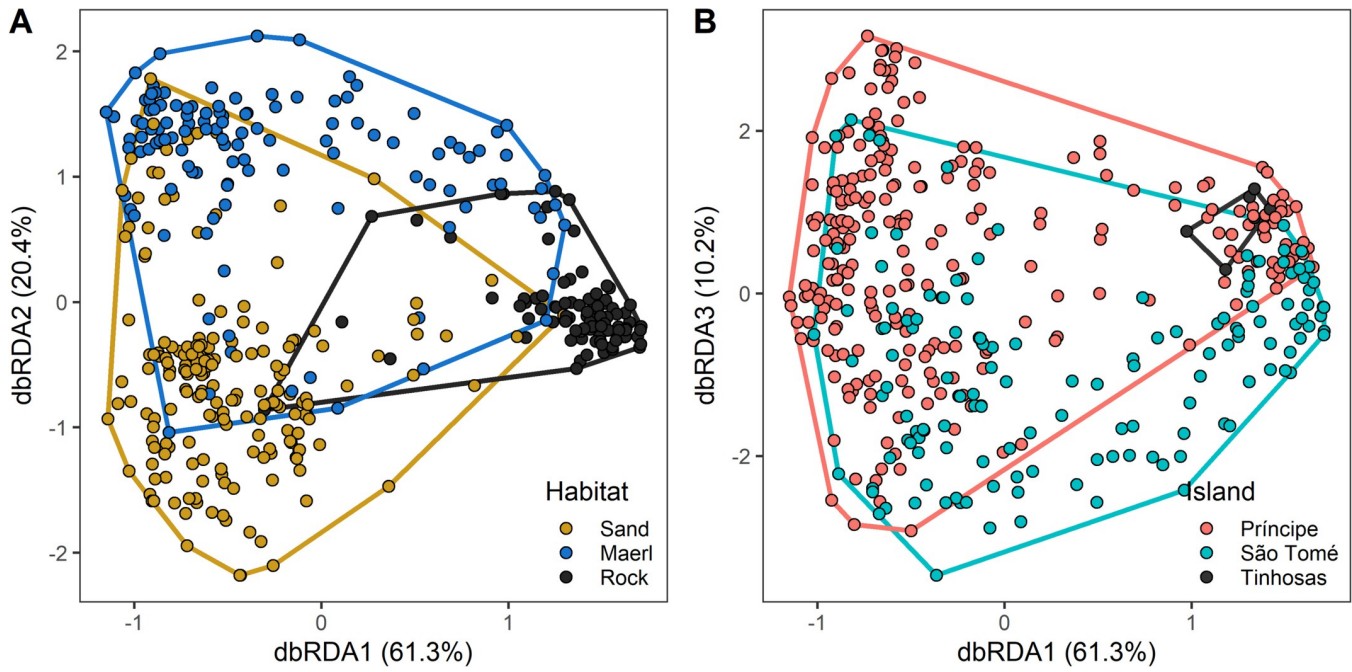

**Fig 4. Distribution of habitats and islands in the space generated by dbRDA1, dbRDA2 and dbRDA3, with ellipses delineating the space occupied by habitats and island factors.**

season (p<0.01). Pairwise comparisons found significant differences between all habitat types and islands including the Tinhosas (p<0.01).

SIMPER pairwise comparisons for habitat (n = 3) indicated that 19 to 24 species contributed to 70% of dissimilarities across habitats, with between-habitat differences of 12 to 19 of these species contributing significantly to overall dissimilarities (SIMPER permutations, p < 0.05). On rocky habitats, higher abundances of reef fish and schooling species (such as Creole fish, brown chromis or Newton's wrasse), as well as commercial species (such as African red snapper), significantly contributed to dissimilarities between rocky and non-rocky habitats (p < 0.01; sand and maerl; Fig 5 and S11 Table). Santomean comber, pearly razorfish (*Xyrichtys novacula*), flying gurnard (*Dactylopterus volitans*) and flatfishes (*Bothus* sp.) were most abundant on sand and maerl, with different abundances of these species significantly driving dissimilarities between these two habitats (p < 0.01; Fig 5 and S11 Table). For example, Santomean comber was exclusively observed in deployments where maerl rhodoliths were present (including sparse rhodoliths on sandy flats), with higher abundances of flying gurnards on maerl, and flatfishes on sand.

SIMPER pairwise comparisons for island levels (n = 3) indicated that 26 to 28 species contributed to 70% of overall dissimilarities across islands, with between-island differences of 5 to 9 of these species contributing significantly to overall dissimilarities (SIMPER permutations, p < 0.05). Higher abundance of blue runner on Príncipe and higher abundance of reef fish and small schooling species on São Tomé (such as Creole fish and Newton's wrasse) were significantly driving dissimilarities between Príncipe and São Tomé Islands (Fig 5). In São Tomé, a higher abundance of orangespotted filefish and smooth puffer (*Lagocephalus laevigatus*) also contributed significantly to dissimilarities (S12 Table). These two species were often observed forming large schools of up to 50 individuals in São Tomé but not in Príncipe. On the Tinhosas, higher abundances of African red snapper, great barracuda (*Sphyraena barracuda*), and

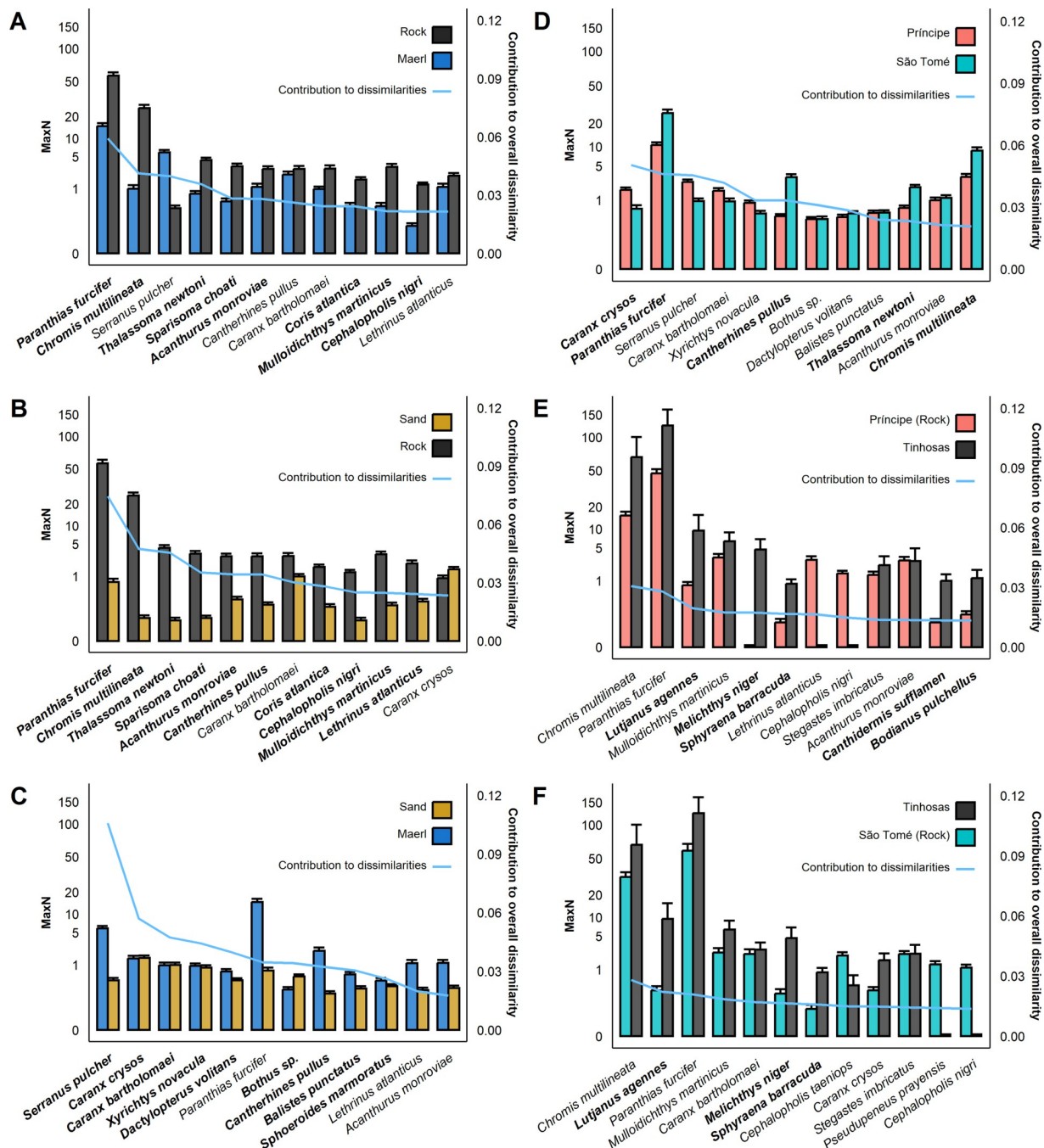

**Fig 5.** Average contribution to dissimilarities (solid blue line) and mean MaxN with standard error (SE) bars of the species that contributed most to overall dissimilarities between habitats (A-C) and islands (D-F), organised from highest (left) to lowest (right) average contribution to overall dissimilarity between pairs (S11 and S12 Tables). To compare between Tinhosas and the other islands (Príncipe and São Tomé), only observations on rocky reefs were used, since this was the only habitat type found at Tinhosas. Highlighted in bold, the species for which differences between pairs of categories contributed significantly (p<0.05) to overall dissimilarities.

black triggerfish, were significantly driving dissimilarities between Tinhosas and the other islands (p < 0.01; Fig 5).

## Discussion

This is the first large-scale study to use BRUVS to examine fish assemblages in coastal waters of a tropical archipelago on the Atlantic coast of Africa. Despite being located in a marine bio-diversity hotspot [13], the archipelago of STP, its marine ichthyofauna, and the environmental and physical factors impacting species assemblages have been relatively understudied [29, 48]. While previous studies have used UVC to characterise drivers of fish distribution on rocky reef habitats [28, 35], to date, this is the first study to use BRUVS to characterise fish communities across main habitat types and to distinguish differences between islands.

### Species assemblages across islands

Data from BRUVS allowed for the identification of 146 species across all islands and habitats. Brown chromis and Creole fish (two small planktivorous species) dominated relative abundances on all islands. This aligns with patterns found on rocky reefs of tropical Atlantic oceanic islands, where small planktivorous fish are the dominant trophic guild and comprise a much higher proportion of the total abundance than continental sites [49]. We also found a relatively low abundance of sharks across the archipelago, with a mean relative abundance of sharks (0.05 sharks per hour) much lower than observed in BRUVS studies across other tropical locations [50], including other tropical oceanic islands in the Atlantic Ocean (e.g., Trindade, 0.32 sharks per hour, [51]). Previous studies using UVC in São Tomé island also found low abundances of sharks and other top predators (such as snappers), potentially related to the effect of long-term fishing pressure [28, 29], and a recent study using interviews indicate that fishers in São Tomé perceive a decline in shark abundance in the last decades [27].

Although species richness and relative abundance were not significantly different between islands, fish assemblages were significantly different. Some of these differences may be related to the higher fishing pressure on São Tomé (12 times higher than on Príncipe) [52], which concentrates within a smaller area of insular shelf. This could explain the lower abundance of blue runner in São Tomé (an important component of the catch of artisanal fisheries) [24], and the seasonal schools of smooth puffer observed in São Tomé–potentially caused by a depletion of top predators, as observed in the Arabian sea [53]. Fishers in São Tomé report that these seasonal schooling events cause significant damage to fishing gears and have been increasing over the last decades [54]. Additionally, abundance of Creole fish and brown chromis was higher in São Tomé. Higher abundance of small planktivorous fish has been related to overfishing effects [55], but can also be related to the influence of upwelling currents [56]. Since Southern São Tomé is reported to be affected by an upwelling [23], further research should focus on understanding how other environmental variables (e.g., potential upwelling currents) may affect species assemblages.

Although the sample size for the Tinhosas was small (n = 6), these deployments comprise, to our knowledge, the first fish surveys conducted in this important area. Our results indicate that the islets host distinct fishing assemblages from the larger islands, with higher abundances of snappers and barracudas potentially reflecting the lower levels of fishing pressure that have been observed around the islets [57]. Additionally, the islets may also have distinct environmental conditions, as shown by the presence and high abundance of black triggerfish, a species that was also observed in the Rolas Islet (Southern São Tomé). This species is normally associated to exposed reef areas [58], as found in the Tinhosas and the Rolas Islet, located at the south-facing shelf edges that result in a steeper slope and higher ocean exposure (Fig 1). Additionally, the high seabird density of the Tinhosas [18] may also be increasing fish biomass through enhanced nutrient supply [59].

## Drivers of fish diversity

Mean species richness in this study was higher than richness observed employing UVC on reef areas on Príncipe [60] and São Tomé [28], which is consistent with other studies comparing both methods [61]. Of the species commonly caught by STP artisanal fisheries (see [24]), benthic BRUVS were effective at sampling demersal or reef-associated species such as snappers (Lutjanidae) or jacks (Carangidae), but other pelagic or mid-water species were not represented in the sample (e.g., *Auxis thazard* and *Euthynnus alleteratus*). Studies comparing benthic BRUVS to UVC also highlight that BRUVS have higher sensitivity to planktonic species but lower sensitivity to small, cryptic, benthic species [61]. These aspects therefore need careful consideration when designing sampling strategies that are appropriate to monitoring goals, and further research should explore other sampling methods, such as mid-water BRUVS (e.g. [8]), to better understand composition of pelagic or mid-water fish communities not captured by benthic BRUVS.

Broad habitat types strongly influenced diversity indicators. Species evenness was lowest on rocky reefs, probably due to the presence of large schools of small reef fish, and species richness and relative abundance were highest on rocky reefs and lowest on sand. This may be reflecting an increasing gradient of habitat complexity across sand, maerl, and rocky reefs, corresponding to increasing richness and relative abundance [62]. Nevertheless, using UVC, Maia et al. [28] found that topographic complexity of reef areas on São Tomé had no effect on species richness or abundance, with other factors (e.g., benthic cover and exposure) having stronger influence. This difference might be explained by the greater range of topographic complexity represented by the habitat classes used in this study, which may elicit a stronger response from fish communities (namely greater richness and abundance on reef areas compared to less structured habitats such as sand and maerl). Additionally, slope, which measured seascape topographic complexity at a higher spatial scale (120 m), had a significant, positive effect on species richness and abundance, with a significant effect on species assemblages across the observed slope range.

We found significant differences in fish community composition across habitat types. On Príncipe Island, Otero-Ferrer et al. [35] found highest species richness on rocky reefs, but highest disparity of taxa in transitions to maerl and sand due to the mixing of biotas [35]. Consistently, our results show that sand, maerl, and rocky reefs supported distinct species assemblages, highlighting the collective role of diverse habitat mosaics on regional ecological processes and biodiversity [62], which should be considered into the design of spatial management measures. For example, rocky reefs had higher abundance of snappers, (an important group for artisanal fisheries), while flatfishes were most abundant on sand. Additionally, maerl beds hosted highly specialist species (e.g., Santomean comber), and had the highest abundance of flying gurnard, a species of local economic and cultural importance targeted by artisanal fisheries [24, 28], whose epipelagic, oceanic larvae constitute the dominant prey of the Tinhosas' seabird population [18]. Indeed, the role of maerl beds as dynamic habitats that comprise productive fishing grounds and nursery areas is well established in several locations in the Atlantic [63]. Thus, while all habitats contributed to overall species richness, rocky reefs and maerl should be considered priority habitats for management, as they host higher abundances of species targeted by artisanal fisheries. Nevertheless, other taxa important for artisanal fisheries (such as blue runner) did not show major differences in abundance across habitats (S11 Table) and, thus, may not be directly affected by zonation based in habitat classes considered in this study.

We found increasing species richness and relative abundance with increasing depth. Two previous studies in STP at a depth range of 3 to 35m obtained two different results on the

effects of depth on reef fish diversity, with Tuya et al. [60] finding a negative effect and Maia et al. [28] finding a positive effect (the latter corroborating our findings). Previous studies conducted elsewhere have associated a similar increasing gradient to a depth refugia effect, since deeper areas may be more sheltered from fishing [64]. However, there is currently no evidence that deeper areas within this range are less exposed to fishing, since fishers in STP regularly fish in areas deeper than 30 m [24, 57]. Peaks in species richness at intermediate depths (30 m) have also been described as the result of species turnover between shallow and deep fish assemblages [65], which might explain our results. Furthermore, Maia et al. [28] suggest that this pattern might result from the presence of a constant thermocline at depths of 20 to 30 m, allowing the local coexistence of species with different environmental affinity.

Our result showed that seasonality affected species richness and evenness. This aligns with BRUVS studies in other tropical locations and highlights the importance of incorporating seasonal effects into sampling design and analysis [e.g., 66]. Our results also showed that seasonal effects on richness and evenness were not consistent across islands, which may be indicative of seasonal factors affecting each island differently (e.g., different upwelling patterns, [23]). Further research should therefore focus on exploring of how other biotic and abiotic variables around the islands (e.g., temperature or salinity) may vary seasonally and across locations and how these may be interacting with fish species, particularly those affected by artisanal fisheries.

## Conclusions

To our knowledge, this is the largest BRUVS survey conducted in the Gulf of Guinea region to date. Our results provide an important benchmark to inform conservation actions and establish monitoring priorities, offering important insights on fish diversity, including on species of conservation concern or cultural and economic importance for coastal communities. The results presented here have implications for informing the ongoing implementation of spatial management strategies (such as MPAs), identifying priority habitats for conservation. Our results also suggest that fish communities in STP may have been impacted by long-term fishing effort, especially on São Tomé Island. Further research should therefore monitor potential impacts of putative threats to support conservation efforts. Additionally, the broader socio-economic context needs to be addressed, as integrating ecological information with stakeholders' resource-use, values and perspectives is critical for conservation success [26]. Finally, the results from this study strengthen our knowledge on fish community ecology and habitat occupancy in STP, providing a benchmark that can serve as a stepping stone to further investigation. This study also provides a general protocol that could potentially be applied to other countries in the Gulf of Guinea region to substantially fill their paucity of ecological data.

## Supporting information

**S1 Fig. Location of Príncipe's deployments.** Map illustrating: (1) sampling area; (2) type of sampling (random / non-random); (3) year (2018/2019 or 209/2020); and (4) season ("gravana", June to September; "summer", December to February). Note that, for Príncipe's 2019 / 2020 "summer" deployments, COVID-19 restrictions forced to delay sampling (March to July 2020), effectively entering the gravana season for that sampling round (Bathymetric data: © GEBCO).
(TIF)

**S2 Fig. Location of São Tomé's deployments.** Map illustrating: (1) sampling area; (2) type of sampling (random / non-random); and (3) season ("gravana", June to September; "summer",

December to February) (Bathymetric data: © GEBCO).
(TIF)

**S3 Fig. Depth and slope of the 417 deployments retained for analysis, disaggregated by island and habitat.**
(TIF)

**S4 Fig. Partial effects of effects in models described in S6 Table, with residuals plotted as points.** Partial effects and residuals were estimated using the R package "gratia" [47].
(TIF)

**S5 Fig. db-RDA ordination plot representing location of deployments across dbRDA axes.** Arrows represent the coefficients of each variable on the constrained dbRDA axes, and crosses and dashed lines represent the centroid of factors in the ordination. The length of the arrow and dashed lines represent the strength of the effect of a variable or factor on community composition, and the direction (equal or opposite) indicates whether variables have positive or negative effects on the variation represented by constrained axes. See also S8 Table containing coefficients and centroids of each variable in the dbRDA ordination.
(TIF)

**S6 Fig.** A) Mean species richness at 5-min intervals was calculated across total soak time to evaluate optimal recording times. Plots show cumulative means of species richness at different soaking times with standard error bars, disaggregated by island and habitat type. Mean species richness increased with soak time duration, with on average, 90% of species observed following 45 minutes of deployment (all habitats and islands combined). B) To assess whether sampling effort (number of sites) accurately captured species richness rarefaction curves were generated by randomly adding sites across 100 permutations using R package vegan (Oksanen et al., 2022). Plots show species accumulation plots for Príncipe and São Tomé where number of sites = BRUV deployments.
(TIF)

**S1 Table. Sampling effort per island and per sampling period.**
(DOCX)

**S2 Table. Summary of environmental conditions of deployments.**
(DOCX)

**S3 Table. Species' occurrences (expressed as counts and as percentages of deployments), total MaxN across deployments, mean MaxN per deployment and mean MaxN per occurrence.**
(DOCX)

**S4 Table. Family's species richness, occurrence (defined as number of deployments and percentage of deployments in which a species is observed), and abundance (defined as MaxN totals and mean MaxN per deployment).**
(DOCX)

**S5 Table. Mean and standard deviation of species richness (S), relative abundance (MaxN) and E1/D evenness index, disaggregated by habitat and island.**
(DOCX)

**S6 Table. Generalised Additive Models (GAMs) to assess the effect of environmental variables (depth, distance to shore, Slope, Habitat, Island and Season) on richness (S), abundance (MaxN), and E1/D evenness index.** Habitat and island were fitted as categorical effects;

depth, distance to shore and slope were fitted as thin plate splines; and season and season by habitat were fitted as cyclic splines. Variable smooths were fitted via Restricted Maximum Likelihood (REML).
(DOCX)

**S7 Table. Degree of support of the effect of environmental variables on diversity indicators, expressed as the summed weights of models in S6 Table.** Values closer to 1 indicate stronger and more consistent support for a predictor across the retained models in the top model set.
(DOCX)

**S8 Table. Contribution of predictors to the deviance explained by each model in the top model set.** Grey-shaded cells indicate that the variable was not included in the model.
(DOCX)

**S9 Table. Results of an ANOVA-like permutation significance test for the effect of each dbRDA constrained axis on response variables, conducted with the function vegan::anova. cca() in R, and percentage of constrained variance explained by each of the constrained axes.**
(DOCX)

**S10 Table. Variable coefficients for each of the constrained dbRDA axes and centroids of factor variables on each of the constrained axes.** Constrained dbRDA axes are orthogonal combinations of the explanatory variables (i.e., a multiple regression model) that best explain, in successive order, the variation of the response matrix (Borcard, Gillet & Legendre, 2011).
(DOCX)

**S11 Table. SIMPER pairwise comparisons of habitat types, with mean MaxN (and SD), and average contribution of each species to overall dissimilarities amongst pairs.** Significance of the contribution of each species to overall dissimilarities between pairs was assessed using permutation tests.
(DOCX)

**S12 Table. SIMPER pairwise comparisons of islands, with mean MaxN (and SD), and average contribution of each species to overall dissimilarities amongst pairs.** Significance of the contribution of each species to overall dissimilarities between pairs was assessed using permutation tests. To compare the Tinhosas to Príncipe and São Tomé, only rocky reefs were considered, since the Tinhosas are a rocky reef environment and sand and maerl are not present.
(DOCX)

**S1 File. Additional methodological information.**
(DOCX)

**S2 File. Summary of deployments and environmental conditions.**
(DOCX)

**S3 File. Inclusivity in global research questionnaire.**
(DOCX)

## Acknowledgments

We thank Fundação Príncipe's marine team for their support during field work, especially marine guards Ibizaltino Ceita, Abduley de Pina, Pascoal Monteiro, Zelinho Maquengo and Carlos Sanches, and project coordinators Vanessa Schmitt and Jaconias Pereira. We also wish

to thank the team that supported São Tomé's sampling, including captain Juvêncio and crew members, as well as Bastien Loloum.

## Author Contributions

**Conceptualization:** Guillermo Porriños, Kristian Metcalfe, Ana Nuno, Katy Walker, Philip D. Doherty.

**Data curation:** Guillermo Porriños.

**Formal analysis:** Guillermo Porriños.

**Funding acquisition:** Ana Nuno, Annette C. Broderick, Brendan J. Godley, Luisa Madruga, Berry Mulligan.

**Investigation:** Guillermo Porriños, Manuel da Graça, Adam Dixon, Márcio Guedes, Lodney Nazaré, Albertino dos Santos, Liliana P. Colman, Jemima Dimbleby, Marta Garcia-Doce, Philip D. Doherty.

**Methodology:** Guillermo Porriños, Kristian Metcalfe, Ana Nuno, Philip D. Doherty.

**Project administration:** Guillermo Porriños, Ana Nuno, Katy Walker, Annette C. Broderick, Luisa Madruga, Berry Mulligan.

**Resources:** Guillermo Porriños, Ana Nuno, Katy Walker, Luisa Madruga.

**Software:** Guillermo Porriños.

**Supervision:** Guillermo Porriños, Kristian Metcalfe, Ana Nuno, Tiago Capela Lourenço, Philip D. Doherty.

**Visualization:** Guillermo Porriños.

**Writing – original draft:** Guillermo Porriños, Kristian Metcalfe, Ana Nuno, Katy Walker, Philip D. Doherty.

**Writing – review & editing:** Guillermo Porriños, Kristian Metcalfe, Ana Nuno, Manuel da Graça, Katy Walker, Adam Dixon, Márcio Guedes, Lodney Nazaré, Albertino dos Santos, Liliana P. Colman, Jemima Dimbleby, Marta Garcia-Doce, Annette C. Broderick, Brendan J. Godley, Tiago Capela Lourenço, Luisa Madruga, Hugulay Albuquerque Maia, Berry Mulligan, Philip D. Doherty.

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
