## [Decision Letter · Decision Letter 0]

6 Sep 2024

PONE-D-24-35103Fish community composition in the tropical archipelago of São Tomé and PríncipePLOS ONE

Dear Dr. Porriños,

Thank you for submitting your manuscript to PLOS ONE. After careful consideration, we feel that it has merit but does not fully meet PLOS ONE’s publication criteria as it currently stands. Therefore, we invite you to submit a revised version of the manuscript that addresses the points raised during the review process.

We look forward to receiving your revised manuscript.

Kind regards,

Tzen-Yuh Chiang

Academic Editor

PLOS ONE

“Fieldwork described here was undertaken within the project “Establishing a network of marine protected areas across São Tomé and Príncipe through a co-management approach” and was funded by Blue Action Fund and Arcadia Fund for Nature - a charitable fund of Lisbet Rausing and Peter Baldwin. Preliminary work was funded by the Darwin Initiative - a UK government grant scheme (Project 23–012), Forever Príncipe and Halpin Trust. G.P. acknowledges funding from Fundação para a Ciência e Tecnologia (FCT) doctoral grant nº UI/BD/151263/2021. A.N. acknowledges the support of the European Union's Horizon 2020 research and innovation programme under the Marie Skłodowska‐Curie grant agreement SocioEcoFrontiers No. 843865. K.M. acknowledges funding support from the Darwin Initiative (Project 23-011 and 26-014), Waterloo Foundation, and the Wildlife Conservation Society Congo Programme. G.P and T.C.L. acknowledge the support received from cE3c - Center for Ecology, Evolution and Environmental Changes through FCT’s strategic project UIDB/00329/2020 (doi: 10.54499/UIDB/00329/2020)”

“Fieldwork described here was undertaken within the project “Establishing a network of marine protected areas across São Tomé and Príncipe through a co-management approach” and was funded by Blue Action Fund and Arcadia Fund for Nature - a charitable fund of Lisbet Rausing and Peter Baldwin. Preliminary work was funded by the Darwin Initiative - a UK government grant scheme (Project 23–012), Forever Príncipe and Halpin Trust. G.P. acknowledges funding from Fundação para a Ciência e Tecnologia (FCT) doctoral grant nº UI/BD/151263/2021. A.N. acknowledges the support of the European Union's Horizon 2020 research and innovation programme under the Marie Skłodowska‐Curie grant agreement SocioEcoFrontiers No. 843865. K.M. acknowledges funding support from the Darwin Initiative (Project 23-011 and 26-014), Waterloo Foundation, and the Wildlife Conservation Society Congo Programme. G.P and T.C.L. acknowledge the support received from cE3c - Center for Ecology, Evolution and Environmental Changes through FCT’s strategic project UIDB/00329/2020 (doi: 10.54499/UIDB/00329/2020). We further thank Fundação Príncipe’s marine team for their support during field work, especially marine guards Ibizaltino Ceita, Abduley de Pina, Pascoal Monteiro, Zelinho Maquengo and Carlos Sanches, and project coordinators Vanessa Schmitt and Jaconias Pereira. We also wish to thank the team that supported São Tomé’s sampling, including captain Juvêncio and crew members, as well as Bastien Loloum.”

“Fieldwork described here was undertaken within the project “Establishing a network of marine protected areas across São Tomé and Príncipe through a co-management approach” and was funded by Blue Action Fund and Arcadia Fund for Nature - a charitable fund of Lisbet Rausing and Peter Baldwin. Preliminary work was funded by the Darwin Initiative - a UK government grant scheme (Project 23–012), Forever Príncipe and Halpin Trust. G.P. acknowledges funding from Fundação para a Ciência e Tecnologia (FCT) doctoral grant nº UI/BD/151263/2021. A.N. acknowledges the support of the European Union's Horizon 2020 research and innovation programme under the Marie Skłodowska‐Curie grant agreement SocioEcoFrontiers No. 843865. K.M. acknowledges funding support from the Darwin Initiative (Project 23-011 and 26-014), Waterloo Foundation, and the Wildlife Conservation Society Congo Programme. G.P and T.C.L. acknowledge the support received from cE3c - Center for Ecology, Evolution and Environmental Changes through FCT’s strategic project UIDB/00329/2020 (doi: 10.54499/UIDB/00329/2020)”

6. We note that Figures 1, S1 and S2 in your submission contain [map/satellite] images which may be copyrighted. All PLOS content is published under the Creative Commons Attribution License (CC BY 4.0), which means that the manuscript, images, and Supporting Information files will be freely available online, and any third party is permitted to access, download, copy, distribute, and use these materials in any way, even commercially, with proper attribution. For these reasons, we cannot publish previously copyrighted maps or satellite images created using proprietary data, such as Google software (Google Maps, Street View, and Earth). For more information, see our copyright guidelines: http://journals.plos.org/plosone/s/licenses-and-copyright.

a. You may seek permission from the original copyright holder of Figures 1, S1 and S2 to publish the content specifically under the CC BY 4.0 license. 

Reviewers' comments:

Reviewer's Responses to Questions

**Comments to the Author**

1. Is the manuscript technically sound, and do the data support the conclusions?

Reviewer #1: Yes

2. Has the statistical analysis been performed appropriately and rigorously? 

Reviewer #1: Yes

3. Have the authors made all data underlying the findings in their manuscript fully available?

Reviewer #1: Yes

4. Is the manuscript presented in an intelligible fashion and written in standard English?

Reviewer #1: Yes

5. Review Comments to the Author

Reviewer #1: This is well-written manuscript describing assemblage of fish at Sao Tome and Pricipe in the Gulf of Guinea. The research approach and findings are not particularly novel, but this represents valuable data in a location where fish data are missing.

The BRUV methodology is executed very well and so are the analyses. I have minor comments and edits.

Line 35: change “whose” to “where coastal…”

Line 70: the writing is generally good, however this sentence was jarring. “Commonly used monitoring techniques to assess the health …” should read “Common techniques used to monitor and assess the health …

Line 108 and 109: you mention large fluctuations in SST but the mean values are 26 and 27 between seasons. This is not large, but I suspect this is because it is a mean. Maybe you could report the range or standard error or just say there are fluctuations in temperature, not necessarily large.

Line 134: the [11] is somewhat outdated. I understand the work was undertaken in 2019/20 but you should use:

Langlois, T., Goetze, J., Bond, T., Monk, J., Abesamis, R.A., Asher, J., Barrett, N., Bernard, A.T.F., Bouchet, P.J., Birt, M.J., Cappo, M., Currey‐Randall, L.M., Driessen, D., Fairclough, D. V., Fullwood, L.A.F., Gibbons, B.A., Harasti, D., Heupel, M.R., Hicks, J., Holmes, T.H., Huveneers, C., Ierodiaconou, D., Jordan, A., Knott, N.A., Lindfield, S., Malcolm, H.A., McLean, D., Meekan, M., Miller, D., Mitchell, P.J., Newman, S.J., Radford, B., Rolim, F.A., Saunders, B.J., Stowar, M., Smith, A.N.H., Travers, M.J., Wakefield, C.B., Whitmarsh, S.K., Williams, J., Harvey, E.S., 2020. A field and video annotation guide for baited remote underwater stereo‐video surveys of demersal fish assemblages. Methods in Ecology and Evolution 11, 1401–1409. https://doi.org/10.1111/2041-210X.13470

This paper is comprehensive and your methods are very well aligned.

Section 2.3 is great to see.

Line 160: analysis is not often started five minutes after landing on the seafloor but this is acceptable. Sometimes this is a period where you find the most cryptic species which are visible before the bait attending specie arrive. Something to consider in the future.

Line 162: use original cite for MaxN Priede papers and Cappo papers. see BRUV paper suggested above for more detail.

Line 165: just call it MaxN not maximum MaxN. MaxN is already the maximum.

Section 2.5: I wonder why you have not used year as a factor too. I understand that you added more deployments after not getting enough reef in the first year, but it would be interesting to see if anything changed between years. Habitat and depth is still going to drive everything.

Statistics: good!

Results: thorough and reporting all the metrics I like to see.

Line 398: change “that” to “than”.

Discussion: good. You have focusses on the ecology, mostly. I would like to see more around the fisheries. I know you have not measured any fish, but I believe you could comment on size classes and if you observed any larger fish deeper or further away.

Fig 3: Depth by Richness RI = 1? These plots look too perfect for a GAM. Having points scattered on the plots would provide the reader with confidence that they are correct. Also, consider plotting your season facet in a polarplot. This joins up each end – see examples in

Bond, T., Langlois, T.J., Partridge, J.C., Birt, M.J., Malseed, B.E., Smith, L., McLean, D.L., 2018. Diel shifts and habitat associations of fish assemblages on a subsea pipeline. Fisheries Research 206, 220–234. https://doi.org/10.1016/j.fishres.2018.05.011

6. PLOS authors have the option to publish the peer review history of their article (what does this mean?). If published, this will include your full peer review and any attached files.

Reviewer #1: No

---

## [Author Response · Author response to Decision Letter 0]

27 Sep 2024

RESPONSE TO EDITOR

Response to COMMENT 1: Format of headings and subheadings, cross references to supplementary materials and captions have been updated to adjust PLOS ONE style templates.

Response to COMMENT 2: We have included a complete copy of PLOS’ questionnaire on inclusivity in global research.

Response to COMMENT 3: Thank you for pointing this out. Funders had no role in the study. 

The Role of Funder statement should read as: 

The full Financial Disclosure statement should read as:

• “Fieldwork described here was undertaken within the project “Establishing a network of marine protected areas across São Tomé and Príncipe through a co-management approach” and was funded by Blue Action Fund and Arcadia Fund for Nature - a charitable fund of Lisbet Rausing and Peter Baldwin. Preliminary work was funded by the Darwin Initiative - a UK government grant scheme (Project 23–012), Forever Príncipe and Halpin Trust. G.P. acknowledges funding from Fundação para a Ciência e Tecnologia (FCT) doctoral grant nº UI/BD/151263/2021. A.N. acknowledges the support of the European Union's Horizon 2020 research and innovation programme under the Marie Skłodowska‐Curie grant agreement SocioEcoFrontiers No. 843865. K.M. acknowledges funding support from the Darwin Initiative (Project 23-011 and 26-014), Waterloo Foundation, and the Wildlife Conservation Society Congo Programme. G.P and T.C.L. acknowledge the support received from cE3c - Center for Ecology, Evolution and Environmental Changes through FCT’s strategic project UIDB/00329/2020 (doi: 10.54499/UIDB/00329/2020). The funders had no role in study design, data collection and analysis, decision to publish, or preparation of the manuscript.”

Response to COMMENT 4: Thanks for pointing this out. Acknowledgements section has been updated accordingly.

Response to COMMENT 5: Thank you for clarification. Both links provided in the data availability statement are active and data has already been uploaded, but it is embargoed until acceptance of the manuscript (https://doi.org/10.5281/zenodo.13326736 and https://github.com/gporrinos/BRUVS_SaoTomePrincipe_2018_2020). Data will be immediately made available upon confirmation of acceptance.

Response to COMMENT 6: Thank you for pointing this out. We had used bathymetry data from two digitised nautical charts from “Instituto Hidrográfico Português”. Nevertheless, since we have not been able to obtain the copyright information for that data, we have substituted that data by the GEBCO bathymetry data in all figures. The following statement has been added to the S1 and S2 Figure legends to reflect the new data source: “(Bathymetric data: © GEBCO)”.

Response to COMMENT 7: We have reviewed the reference list and have not found any reference that has been retracted.

RESPONSE TO REVIEWER 1'S COMMENTS

COMMENT 1. This is well-written manuscript describing assemblage of fish at Sao Tome and Pricipe in the Gulf of Guinea. The research approach and findings are not particularly novel, but this represents valuable data in a location where fish data are missing.

The BRUV methodology is executed very well and so are the analyses. I have minor comments and edits.

RESPONSE 1: Thank you!

COMMENT 2. Line 35: change “whose” to “where coastal…”

RESPONSE 2: Thanks for the suggestion, text was changed accordingly.

COMMENT 3. Line 70: the writing is generally good, however this sentence was jarring. “Commonly used monitoring techniques to assess the health …” should read “Common techniques used to monitor and assess the health …

RESPONSE 3: Thanks for the suggestion, text was changed accordingly.

COMMENT 4. Line 108 and 109: you mention large fluctuations in SST but the mean values are 26 and 27 between seasons. This is not large, but I suspect this is because it is a mean. Maybe you could report the range or standard error or just say there are fluctuations in temperature, not necessarily large.

RESPONSE 4: Thank you for pointing this out! These values were a mistake, the actual values reported in the reference [19] were above 28 and below 23. This has been corrected in the main text.

COMMENT 5. Line 134: the [11] is somewhat outdated. I understand the work was undertaken in 2019/20 but you should use:

Langlois, T., Goetze, J., Bond, T., Monk, J., Abesamis, R.A., Asher, J., Barrett, N., Bernard, A.T.F., Bouchet, P.J., Birt, M.J., Cappo, M., Currey‐Randall, L.M., Driessen, D., Fairclough, D. V., Fullwood, L.A.F., Gibbons, B.A., Harasti, D., Heupel, M.R., Hicks, J., Holmes, T.H., Huveneers, C., Ierodiaconou, D., Jordan, A., Knott, N.A., Lindfield, S., Malcolm, H.A., McLean, D., Meekan, M., Miller, D., Mitchell, P.J., Newman, S.J., Radford, B., Rolim, F.A., Saunders, B.J., Stowar, M., Smith, A.N.H., Travers, M.J., Wakefield, C.B., Whitmarsh, S.K., Williams, J., Harvey, E.S., 2020. A field and video annotation guide for baited remote underwater stereo‐video surveys of demersal fish assemblages. Methods in Ecology and Evolution 11, 1401–1409. https://doi.org/10.1111/2041-210X.13470

This paper is comprehensive and your methods are very well aligned.

RESPONSE 5: Thanks for providing this reference. We have substituted the Whitmarsh et al (2017) reference by this one in all instances.

COMMENT 6. Section 2.3 is great to see.

RESPONSE 6. Thanks!

COMMENT 7. Line 160: analysis is not often started five minutes after landing on the seafloor but this is acceptable. Sometimes this is a period where you find the most cryptic species which are visible before the bait attending specie arrive. Something to consider in the future.

RESPONSE 7. Thanks for the suggestion! We used a 5-min buffer to avoid interferences with the boat and to account for sediment disturbed following deployment, but we will certainly consider your point on cryptic species observations in the future.

COMMENT 8. Line 162: use original cite for MaxN Priede papers and Cappo papers. see BRUV paper suggested above for more detail.

RESPONSE 8. Thanks for providing the references. They have been changed accordingly:

32. Priede IG, Bagley PM, Smith A, Creasey S, Merrett NR. Scavenging deep demersal fishes of the Porcupine Seabight, north-east Atlantic: observations by baited camera, trap and trawl. J Mar Biol Assoc U K. 1994;74: 481–498. doi:10.1017/S0025315400047615

33. Cappo M, Speare P, De’ath G. Comparison of baited remote underwater video stations (BRUVS) and prawn (shrimp) trawls for assessments of fish biodiversity in inter-reefal areas of the Great Barrier Reef Marine Park. J Exp Mar Biol Ecol. 2004;302: 123–152. doi:10.1016/j.jembe.2003.10.006

COMMENT 9. Line 165: just call it MaxN not maximum MaxN. MaxN is already the maximum.

RESPONSE 9. Thanks for the suggestion, text has been changed accordingly.

COMMENT 10. Section 2.5: I wonder why you have not used year as a factor too. I understand that you added more deployments after not getting enough reef in the first year, but it would be interesting to see if anything changed between years. Habitat and depth is still going to drive everything.

RESPONSE 10. Thanks for the suggestion. We did consider including year as a factor as we agree it would be interesting to investigate seasonal effects. However, given the unbalanced sample (sampling was conducted for two years in Príncipe, but only one year in São Tomé), we decided that our data were not sufficient to make robust conclusions about inter-annual seasonality. Nevertheless, we agree that it would be an important factor to consider, and we would be keen on including it in future analyses (sampling described in this article was the first of an ongoing monitoring of STP’s marine assemblages, which will be conducted every three years).

COMMENT 11. Statistics: good!

Results: thorough and reporting all the metrics I like to see.

RESPONSE 11. Thanks!

COMMENT 12. Line 398: change “that” to “than”.

RESPONSE 12. Thanks for spotting this, text has been changed accordingly.

COMMENT 13. Discussion: good. You have focusses on the ecology, mostly. I would like to see more around the fisheries. I know you have not measured any fish, but I believe you could comment on size classes and if you observed any larger fish deeper or further away.

RESPONSE 13. Thanks! 

Whilst we agree this would be a valuable addition to the discussion, we don’t feel confident in making too many suggestive descriptions of the individuals seen, give that we used mono-BRUVS and therefore can’t estimate sizes. For you interest, we are also currently working on an additional fisheries landing paper that touches on these and other aspects, comparing catch across gears and islands. 

COMMENT 14. Fig 3: Depth by Richness RI = 1? These plots look too perfect for a GAM. Having points scattered on the plots would provide the reader with confidence that they are correct. Also, consider plotting your season facet in a polarplot. This joins up each end – see examples in

Bond, T., Langlois, T.J., Partridge, J.C., Birt, M.J., Malseed, B.E., Smith, L., McLean, D.L., 2018. Diel shifts and habitat associations of fish assemblages on a subsea pipeline. Fisheries Research 206, 220–234. https://doi.org/10.1016/j.fishres.2018.05.011

RESPONSE 14. Thanks for your comments! Responses below

We defined relative importance (RI) presented in the figures as the sum of the weighted AICs of the models in which the predictor of interest was present, which was calculated using the MuMIn::sw function (formerly called MuMIn::importance). This measures the relative importance of a predictor within the top model set. We however acknowledge that this terminology may be confusing, as “relative importance” also refers to the contribution of an individual predictor to the deviance explained by a model. To better reflect our methods, we have substituted “relative importance” by “summed weights” (SW). 

Lines 162 to 168 now read as “Weighted averages of best performing models were used to estimate effects, substituting by zero when a predictor was not present in a model. The relative importance of predictors in the top model set was calculated by summing weighted AICs (Summed Weights, SW) across all models containing the variable of interest (e.g., if, in a top model set, predictor “A” is present in two models with weighted AICs = 0.36 and 0.37, then SWA = 0.73). SW ranges from 0 to 1, with values closer to one indicating higher support and values closer to zero indicating lower support”. Additionally, section “Species richness and relative abundance” of the results has been reworded to better reflect our method.

Additionally, we have calculated the contribution of predictors to the deviance explained by the individual models in the top model set, as we acknowledge that this is an important metric that we had not included in the original version. We have calculated this after Lai et al (2024) (https://doi.org/10.1016/j.pld.2024.06.002). 

Lines 168 to 170 now read as “The contribution of predictors to the deviance explained by the individual models in the model set was calculated after Lai et al [38] to account for the average shared variance of predictors with concurvity”. Results have been included in Table S8 of supplementary materials and, when relevant, have been included in section “Species richness and relative abundance” of results. 

Regarding depth, slope and distance to shore: 

Thanks for noting this. In the previous version, depth and slope were fitted as linear effects (not splines) due to initial data interrogation suggesting such relationships between these variables. However, we acknowledge that we relied on assumptions, and not formal testing. Therefore, we have refitted the models with these variables included as thin plate splines. Depth and slope still show a linear effect (which we have noted in the main text), but distance to shore showed a non-linear effect in the relative abundance models.

Additionally, we have plotted the partial effect of smooths and residuals in S4 Fig, which is referenced in the caption of Fig S3 (line 278).

Thanks for the suggestion on the polar plots. We chose to keep the presentation of the results as they are, as these are most likely to be better interpreted by in-country stakeholders.

---

## [Decision Letter · Decision Letter 1]

15 Oct 2024

Fish community composition in the tropical archipelago of São Tomé and Príncipe

PONE-D-24-35103R1

Dear Dr. Porriños,

We’re pleased to inform you that your manuscript has been judged scientifically suitable for publication and will be formally accepted for publication once it meets all outstanding technical requirements.

Kind regards,

Tzen-Yuh Chiang

Academic Editor

PLOS ONE

Additional Editor Comments (optional):

Reviewers' comments:

Reviewer's Responses to Questions

**Comments to the Author**

1. If the authors have adequately addressed your comments raised in a previous round of review and you feel that this manuscript is now acceptable for publication, you may indicate that here to bypass the “Comments to the Author” section, enter your conflict of interest statement in the “Confidential to Editor” section, and submit your "Accept" recommendation.

Reviewer #1: All comments have been addressed

2. Is the manuscript technically sound, and do the data support the conclusions?

Reviewer #1: (No Response)

3. Has the statistical analysis been performed appropriately and rigorously? 

Reviewer #1: (No Response)

4. Have the authors made all data underlying the findings in their manuscript fully available?

Reviewer #1: (No Response)

5. Is the manuscript presented in an intelligible fashion and written in standard English?

Reviewer #1: (No Response)

6. Review Comments to the Author

Reviewer #1: (No Response)

7. PLOS authors have the option to publish the peer review history of their article (what does this mean?). If published, this will include your full peer review and any attached files.

Reviewer #1: No

---

## [Editor Report · Acceptance letter]

23 Oct 2024

PONE-D-24-35103R1 

PLOS ONE

Dear Dr. Porriños, 

I'm pleased to inform you that your manuscript has been deemed suitable for publication in PLOS ONE. Congratulations! Your manuscript is now being handed over to our production team.

Kind regards, 

on behalf of

Dr. Tzen-Yuh Chiang 

Academic Editor

PLOS ONE